# Lung Ultrasound in the Early Diagnosis and Management of the Mild Form of Meconium Aspiration Syndrome: A Case Report

**DOI:** 10.3390/diagnostics13040719

**Published:** 2023-02-14

**Authors:** Alessandro Perri, Simona Fattore, Giorgia Prontera, Maria Letizia Patti, Annamaria Sbordone, Milena Tana, Vito D’Andrea, Giovanni Vento

**Affiliations:** Department of Woman and Child Health and Public Health, Fondazione Policlinico Universitario Agostino Gemelli IRCCS, 00168 Rome, Italy

**Keywords:** neonatal respiratory distress, meconium aspiration syndrome, lung ultrasound

## Abstract

MAS is a common cause of neonatal respiratory distress in term and post-term neonates. Meconium staining of the amniotic fluid occurs in about 10–13% of normal pregnancies, and about 4% of these infants develop respiratory distress. In the past, MAS was diagnosed mainly on the basis of history, clinical symptoms, and chest radiography. Several authors have addressed the ultrasonographic assessment of the most common respiratory patterns in neonates. In particular, MAS is characterised by a heterogeneous alveolointerstitial syndrome, subpleural abnormalities with multiple lung consolidations, characterised by a hepatisation aspect. We present six cases of infants with a clinical history of meconium-stained fluid who presented with respiratory distress at birth. Lung ultrasound allowed the diagnosis of MAS in all the studied cases, despite the mild clinical picture. All children had the same ultrasound pattern with diffuse and coalescing B-lines, pleural line anomalies, air bronchograms, and subpleural consolidations with irregular shapes. These patterns were distributed in different areas of the lungs. These signs are specific enough to distinguish between MAS and other causes of neonatal respiratory distress, allowing the clinician to optimise therapeutic management.

## 1. Introduction

Respiratory distress is one of the most common diseases in neonatology, affecting the neonate in the first few hours of life. About 7–10% of newborns require respiratory support at birth, with up to 1% requiring extensive resuscitation [1,2,3]. Clinical symptoms include tachypnoea, nasal flaring, grunting, retractions (subcostal, intercostal, supracostal, jugular), and cyanosis. Furthermore, more severe forms can present with apnoea, bradypnoea, irregular breathing, inspiratory stridor, wheezing, and hypoxia [4,5,6]. The most common causes of respiratory distress in neonates are pulmonary in origin and include transient neonatal tachypnoea (TTN), respiratory distress syndrome (RDS), meconium aspiration syndrome (MAS), pneumonia, sepsis, pneumothorax, persistent pulmonary hypertension of the newborn, and delayed transition [4,7]. MAS is a common cause of neonatal respiratory distress in term and post-mature neonates. Meconium staining of the amniotic fluid occurs in about 10–13% of normal pregnancies, and about 4% of these infants develop respiratory distress. Infants born with meconium-stained amniotic fluid (MSAF) are 100 times more likely to develop respiratory distress in the neonatal period than infants born with clear amniotic fluid, even in the absence of prenatal foetal heart rate abnormalities or the need for neonatal resuscitation [8,9,10]. Clinical criteria defining MAS are: (1) respiratory distress (tachypnoea, retractions, or grunting) in a newborn born through meconium-stained amniotic fluid (MSAF); (2) a need for supplemental oxygen to maintain haemoglobin oxygen saturation (SaO2) at 92% or greater; (3) a need for oxygen beginning in the first 2 h of life and continuing for at least 12 h; and (4) the absence of congenital malformations of the airway, lungs, or heart [11,12,13,14].

Depending on the amount and viscosity of the aspirated meconium, respiratory findings range from mild/moderate respiratory distress to severe refractory hypoxemia due to persistent pulmonary hypertension (PPHN), requiring advanced respiratory support (such as high-frequency oscillatory ventilation, inhaled nitric oxide, and extracorporeal membrane oxygenation) [15]. In the past, MAS was diagnosed mainly on the basis of history, clinical symptoms, and chest radiography; ultrasound sonography was not used as a diagnostic tool. However, in recent years, ultrasonography has been successfully used to diagnose many types of lung disease, such as neonatal respiratory distress syndrome (RDS), transient tachypnoea of the newborn, pneumonia, and atelectasis. Lung ultrasound (LUS) has many advantages: it can be easily performed and is suitable for dynamic monitoring; it does not use radiation and therefore prevents radiation-related side effect, for the patient, other patients, and medical staff on the same ward. LUS is easy to learn and has low intra-observer and inter-observer variability when a standardised approach is used [16,17,18,19]. The examination is repeatable; it can be done several times without side effects, providing continuous information regarding the clinical progress of the underlying pathology and making it possible to manage clinical care on the basis of the patient’s response. Furthermore, lung ultrasound allows a quick identification and management of any complications of the respiratory distress of the neonate.

Several authors have addressed the ultrasonographic assessment of the most common respiratory patterns in neonates, such as RDS, TTN, and MAS. Lung ultrasonography seems to be more accurate than chest radiography in identifying these disorders and providing useful information on severity and prognosis. Considering this and the previously listed advantages, its diffusion in Neonatal Intensive Care Units (NICU) is progressively increasing, in parallel with the number of clinicians able to perform it. [20,21,22,23,24,25,26,27]

MAS is characterised by a heterogeneous alveolo-interstitial syndrome, subpleural abnormalities with multiple lung consolidations, and a hepatisation aspect [28,29,30]. Sonographic features of the MAS have been described in the past. Main lung ultrasonographic findings in patients with MAS are consolidation with air bronchogram associated to alveolar-interstitial syndrome or B-line in the non-consolidation area, pleural line anomalies, A-line disappearance, atelectasis, and pleural effusion [27,28].

We present six cases of infants with a clinical history of meconium-stained fluid who presented with respiratory distress at birth. We also discuss the role of the LUS in the management of this disease.

## 2. Materials and Methods

We examined six term infants who had aspirated meconium-stained fluid with respiratory distress at birth and were admitted to the Fondazione Policlinico Universitario ‘‘A. Gemelli’’ IRCCS (Rome, Italy). We included only non-invasively ventilated patients with a diagnosis of MAS. A lung ultrasound was performed by a trained physician within the first 6 h after birth. A 12 MHz linear probe was used. Images were taken with a General Electric LOGIQ E ultrasound machine. The infants were placed in the supine, lateral, or prone position. During the ultrasound examination, we scanned the anterior, lateral, and posterior chest walls, using the anterior and posterior axillary lines as boundaries. When scanning each lung region, the probe was held perpendicular or parallel to the ribs. Non-pharmacological measures, such as non-nutritive sucking and gentle physical containment, were used to prevent patient agitation. All examinations were performed using sterile disposable probe covers, as established by internal protocols.

Demographic and LUS characteristics were collected from electronic patient records (Digistat ^®^) and included gestational age, birth weight, APGAR score, mode of delivery, maternal medical conditions, type and duration of ventilation, O2 requirement, and LUS pattern. We also collected maternal data about gestational diabetes and other gestational pathologies. Data about follow-up were also collected; the neonates were evaluated post-discharge every 2 weeks until 3 months of age or until normalisation of ultrasound signs.

Written informed consent was obtained from parents or guardians for all infants.

## 3. Cases Study

The six observed patients were term neonates with a mean gestational age (GA) of 40 ± 1 weeks; mean of birth weight (BW) was 3280 ± 380 g. Of these patients, five (83%) were born via vaginal delivery, and their weight was appropriate for gestational age (AGA). The median Apgar index was eight at 1 min and nine at 5 min.

The neonates had respiratory distress of variable severity: each required oxygen with a mean requirement of 30% for a mean of 33 h. Only two (33%) neonates needed non-invasive respiratory support with nasal cannula continuous positive airway pressure (nCPAP), with a mean pressure equal to 6 cmH2O for 18 and 24 h. None of them required invasive mechanical ventilation or exogenous surfactant administration.

At the lung ultrasound performed in the first 6 h of life, all the neonates presented the same characteristics, with alveolar-interstitial syndrome shown by a diffuse B-line, often coalescing. All the neonates observed also presented pleural line anomalies such as thickening and interruption (Figure 1). In all the cases, we observed the disappearance of A-lines in the involved areas. Lastly, all of them had subpleural consolidations with irregular shapes, variable sizes, and air bronchograms.

The areas of consolidation were in all cases bilateral and variously distributed in different areas of the lungs (Figure 2), with greater involvement of the right lung in four neonates (66%). In the two patients who required nCPAP, the consolidations were larger and more extensive. Three neonates also underwent chest radiography that showed “diffuse bilateral interstitial involvement” or “diffuse bilateral alveolar-interstitial involvement”.

Data for each infant are shown in Table 1.

## 4. Discussion

We presented six cases of MAS, all of whom did not need invasive ventilation. MAS is a common cause of neonatal respiratory distress in term and post-mature neonates. In the past, this disease was only diagnosed based on history, clinical symptoms, and chest radiography. Ultrasound sonography is a recent diagnostic tool.

The typical X-ray pattern of this syndrome, characterised by lung hyperinflation with cottony and patchy infiltrates alternating with areas of hypertransparency, can identify severe forms of MAS, but it seems unable to recognise mild cases such as those described.

The severity of MAS is directly proportional to the extent of lung involvement as well as the thickness of the meconium. It is likely that in those forms where the amount of amniotic fluid prevails over the inhaled meconium, the viscosity is reduced, leading to the development of milder forms such as those described in this article [30]. In these forms of MAS, a non-specific radiological picture with signs of interstitial or alveolar-interstitial involvement is more common, not allowing the distinction between the different causes of respiratory distress in the newborn, such as RDS and TTN.

Otherwise, in all the studied cases, lung ultrasound allowed the diagnosis of MAS despite the mild clinical picture. All the children had the same ultrasound pattern with diffuse and coalescing B-lines, pleural line anomalies, air bronchograms, and subpleural consolidations of irregular shapes and variable sizes distributed variously in the lungs. These signs are specific enough to distinguish between MAS and other causes of neonatal respiratory distress, allowing the clinician to optimise therapeutic management.

The infants with the most severe ultrasound images presented with more severe pathology, necessitating non-invasive respiratory support. Those with a milder ultrasound picture, instead, developed a milder clinical course, with only the need for oxygen therapy and a shorter duration of symptoms. This suggests that the extent of the consolidations measured by ultrasound is correlated with the severity of the disease, although further studies are needed to verify this finding.

In conclusion, in the management of a newborn with a history and clinic signs compatible with MAS, lung ultrasound seems to be able to identify this condition. Performed in the first hours of life, the LUS exam provides information about the localisation of the most involved areas and about the state of the remaining lung parenchyma. In addition, it is a safe tool that does not use ionising radiation and can be repeated several times. Being a dynamic, harmless, and repeatable exam, LUS allows the neonatologists to evaluate the evolution of the pathology over time and the response to the therapy.

Surely, further studies are needed to confirm our finding in a larger number of newborns, but ultrasound data, together with the clinical history and clinical evaluation, represent a useful tool to optimise the management of these neonates in accordance with the available evidence [16,17,18,20].

In our experience, lung ultrasound has been used to monitor these patients even after clinical remission (at the diagnosis, weekly if on mechanical ventilation, and at the resolution of the symptoms). We observed that, in all six children studied, the consolidations persisted after the resolution of the symptoms, despite clinical well-being. Our proposal is that in the presence of a neonate with a mild form of meconium aspiration syndrome (suspected on a clinical, anamnestic, and ultrasound basis), at the time of resolution of the clinical picture (with stability of vital parameters and good respiratory dynamics, reassuring clinic), it is not necessary to wait for the ultrasound to normalise before discharge from the Neonatal Intensive Care Unit (NICU). The child can be eventually re-evaluated on an outpatient basis, repeating the examination every 2 weeks until normalisation. In our case series, all six neonates had normal lung ultrasound, without consolidation or interruption of the pleural line, within 4 weeks of life. This strategy appears safe and could reduce the length of stay, with favorable effects on costs, parental anxiety, and child well-being.

## Figures and Tables

**Figure 1 diagnostics-13-00719-f001:**
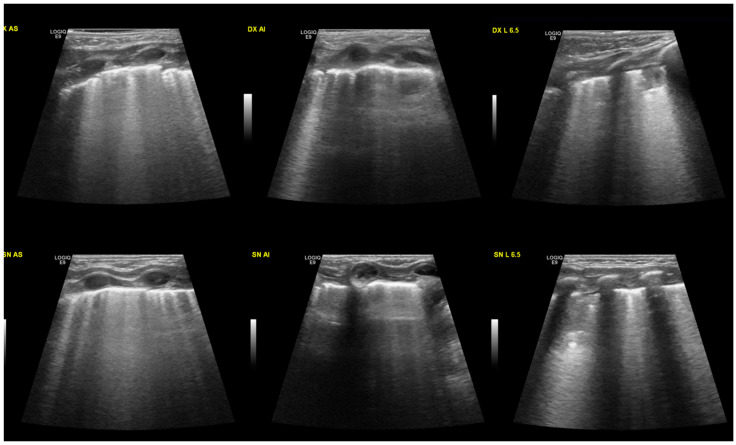
Typical ultrasound findings in MAS: B lines, often coalescing, and the pleural line is thickened and interrupted. Small consolidations (less than 1 cm) can be found in different areas of the lungs. At the top of the image, the right lung is shown, divided into three areas: the upper anterior, lower anterior, and lateral. In the lower portion of the image, the left lung.

**Figure 2 diagnostics-13-00719-f002:**
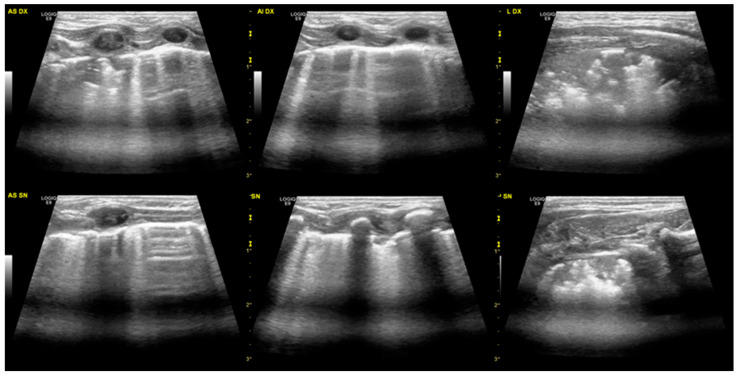
Sub-pleural extended consolidation. This patient has a more severe disease than the one presented in Figure 1. At the top of the image, the right lung is shown, divided into three areas: the upper anterior, lower anterior, and lateral. In the lower portion of the image, the left lung.

**Table 1 diagnostics-13-00719-t001:** Summary of neonatal characteristics. GA: gestational age. BW: birth weight. CTG: cardiotocography. NICU: neonatal intensive care unit.

	Patient 1	Patient 2	Patient 3	Patient 4	Patient 5	Patient 6
**GA**	39 + 6	40 + 2	39 + 6	39 + 3	41 + 0	39 + 4
**BW** **(g)**	3590	3330	2965	2840	3100	3835
**Delivery**	Vaginal delivery	Vaginal delivery	Vaginal delivery	Vaginal delivery	Cesarean section (CTG type 2)	Vaginal delivery
**Maternal desease**	Gestational diabetes	Gestational diabetes	-	-	-	-
**Apgar index 1′-5′**	8–9	9–9	7–8	8–9	5–8	9–10
**Delivery room assistance**	-	-	FiO_2_ 0.40	FiO_2_ 0.35	CPAP 6 cmH_2_O, FiO_2_ 0.80	-
NICU respiratory assistance	-	-	CPAP 6 cmH_2_O	-	-	CPAP 6 cmH_2_O
Hours	-	-	18	-	-	24
FiO_2_	0.27	0.25	0.28	0.30	0.30	0.40
Hours	48	48	24	24	28	24
**CO_2_ (mmHg)**	39	51	38	46	41	30
**Lung ultrasound:**						
Diffuse coalescing B-linee	**✔**	**✔**	**✔**	**✔**	**✔**	**✔**
Thickening/interruption of pleural line	**✔**	**✔**	**✔**	**✔**	**✔**	**✔**
Subpleural consolidation	**✔**	**✔**	**✔**	**✔**	**✔**	**✔**
Largest involvement of the right lung	**✔**	**✗**	**✔**	**✗**	**✔**	**✔**
**Time of normalisation of lung ultrasound**	3.5 weeks	2 weeks	2.5 weeks	4 weeks	2 weeks	4 weeks
**Chest radiography**	diffuse bilateral interstitial involvement	-	diffuse bilateral alveolar-interstitial involvement	-	-	diffuse bilateral interstitial involvement

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
