# Peer review of "Lung Ultrasound in the Early Diagnosis and Management of the Mild Form of Meconium Aspiration Syndrome: A Case Report"

_diagnostics, 2023, doi:10.3390/diagnostics13040719_

Round 1
Reviewer 1 Report
Introduction:
The introduction is well organized and presents relevant information for the content of the article
Case studies:
On line 118 you mentioned that two newborns needed noninvasive respiratory support nCPAP for 18 and 24 hours, but in Table 2 a number of 24 hours is mentioned in both cases. Maybe I didn't understand myself well.
Chest X-ray was performed only in 3 cases. Why did you consider it necessary to perform the chest X-ray in these cases?
Discussion:
At what time was the lung ultrasound repeated during hospitalization in the 6 cases? You mentioned that after discharge it was repeated at 2-week intervals.
It is a well-written article, useful for the management of the mild forms of MAS, nevertheless, minor revisions are needed before acceptance.
Author Response
Case studies:
- On line 118 you mentioned that two newborns needed noninvasive respiratory support nCPAP for 18 and 24 hours, but in Table 2 a number of 24 hours is mentioned in both cases. Maybe I didn't understand myself well.
Thank you for noticing it. It was a typo. We corrected both the manuscript and the table.
- Chest X-ray was performed only in 3 cases. Why did you consider it necessary to perform the chest X-ray in these cases?
Not all the neonatologist perform LUS. If the sonographer is not immediately available the chest x ray is requested as internal protocol requires.
- Discussion: At what time was the lung ultrasound repeated during hospitalization in the 6 cases? You mentioned that after discharge it was repeated at 2-week intervals.
Thank you for the suggestion; we added the information.
Reviewer 2 Report
Basically, this study is well-designed and comprehensively written about the findings of lung USG from mild MAS infants. I largely agree with the points that the authors have made. However, several issues should be addressed.
First, the manuscript on lines 66 to 86 mentions previous studies, but I don't think each study should take each paragraph. I suggest the authors combine them into one paragraph. Moreover, in line 72, "In particular" seems not to be appropriate considering its context.
Second, on page 4, the legend of Figure 1 is not appropriate; "Figure 1 shows" should be avoided and it should give a more detailed description of the findings for readers. Moreover, I suggest pictures in Figures 1 and 2 should present indicate specifically characteristic lung USG findings. For example, what is Diffuse B-line, coalescing thickening and interruption of pleural lines, multiple subpleural constructions, etc. (If the authors could reduce the numbers of USG pictures as possible). Besides, Figure 2 has 7 USG pictures with variable sizes: these busy pictures fail to deliver what the authors have initially aimed to hand over. The legend of Figure 2. should be amended in a similar way to one of Figure 1.
On the other hand, Some information is duplicated in both Tables 1 and 2, and therefore, I suppose it would be okay if Table 1 is removed. By the way, I found that patient No. 5 received oxygens for only 8 hours, which is not applicable to the diagnostic criteria (more than 24 hours) of MAS. Another point that should be improved is the Lung USG findings presented in Table 2. Descriptions are too busy and not effectively delivered to readers; I recommend to show with the presence or absence (with several checking markers) of each characteristic feature plus features typical for RDS or TTN. It could deliver all messages at a glance.
Finally, in lines 175 to 177, "it is reasonable... to ... this condition", in my opinion, the authors could not proclaim this based on the data the authors have presented.
Overall, this script is very educative and can add valuable knowledge to the academy if it is accordingly modified.
Author Response
- First, the manuscript on lines 66 to 86 mentions previous studies, but I don't think each study should take each paragraph. I suggest the authors combine them into one paragraph. Moreover, in line 72, "In particular" seems not to be appropriate considering its context.
We made the suggested correction.
- Second, on page 4, the legend of Figure 1 is not appropriate; "Figure 1 shows" should be avoided and it should give a more detailed description of the findings for readers. Moreover, I suggest pictures in Figures 1 and 2 should present indicate specifically characteristic lung USG findings. For example, what is Diffuse B-line, coalescing thickening and interruption of pleural lines, multiple subpleural constructions, etc. (If the authors could reduce the numbers of USG pictures as possible). Besides, Figure 2 has 7 USG pictures with variable sizes: these busy pictures fail to deliver what the authors have initially aimed to hand over. The legend of Figure 2. should be amended in a similar way to one of Figure 1.
We modified the caption making it more informative. However, we prefer to keep the whole sonogram. We think that the key message (coalescent b line and small consolidation coexistent in different areas of the lungs can be better explained to the reader this way).
- On the other hand, Some information is duplicated in both Tables 1 and 2, and therefore, I suppose it would be okay if Table 1 is removed.
Thank you for the suggestion. We modified the manuscript as requested.
- By the way, I found that patient No. 5 received oxygens for only 8 hours, which is not applicable to the diagnostic criteria (more than 24 hours) of MAS
Thank you for noticing it, we cheeked the database and corrected the manuscript accordingly.
- Another point that should be improved is the Lung USG findings presented in Table 2. Descriptions are too busy and not effectively delivered to readers; I recommend to show with the presence or absence (with several checking markers) of each characteristic feature plus features typical for RDS or TTN. It could deliver all messages at a glance.
We agree with the suggestion. We modified the table 2.
- Finally, in lines 175 to 177, "it is reasonable... to ... this condition", in my opinion, the authors could not proclaim this based on the data the authors have presented.
We modified the manuscript as requested.
Reviewer 3 Report
This reads in my opinion rather as a confirming case series, confirming that ultrasound can be used to diagnose and perhaps quantify to a certain extent MAS, so that i assessed the novelty as average.
The overarching conclusion that this specific enough to distinguish between MAS and other causes of respiratory distress is rather strong, as this is only a case series of cases classified as having MAS, and likely imaging was not ‘disconnected’ from other pieces of clinical information. I therefore would highly recommend to be somewhat less ‘confident’ on this.
Discussion
I do believe that no newborn, and likely any patient, benefits from ‘aggressive intensive care’, so suggest to rephrase this.
Just an open question, but does this journal also allow/enables the upload of video’s, as this would be really beneficial for the ultrasound images.
Author Response
- The overarching conclusion that this specific enough to distinguish between MAS and other causes of respiratory distress is rather strong, as this is only a case series of cases classified as having MAS, and likely imaging was not ‘disconnected’ from other pieces of clinical information. I therefore would highly recommend to be somewhat less ‘confident’ on this.
Thank you with the suggestion, we modified the manuscript as requested.
- Discussion I do believe that no newborn, and likely any patient, benefits from ‘aggressive intensive care’, so suggest to rephrase this.
We agree with you, and we modified the manuscript accordingly.
- Just an open question, but does this journal also allow/enables the upload of video’s, as this would be really beneficial for the ultrasound images.
We are asking the editor if it is possible to add some e-supplemental material. We could add a sonogram providing video clips.
Round 2
Reviewer 2 Report
line 74, "of the mas have been described" --> " ... MAS have been ... "
Figure 1 and 2. I understand what the authors said and respect their opinion. Then I would kindly suggest that every six pictures in both figures should be specified as which part of the lung is indicated. (i.e. Right Upper Lobe, Left Ligular lobe, etc.)
Author Response
line 74, "of the mas have been described" --> " ... MAS have been ... "
Thank you for the suggestion, we changed “mas” with “MAS”.
Figure 1 and 2. I understand what the authors said and respect their opinion. Then I would kindly suggest that every six pictures in both figures should be specified as which part of the lung is indicated. (i.e. Right Upper Lobe, Left Ligular lobe, etc.)
Thank you for your suggestion, we added this information in the caption.